# A Multiepitope Nanovaccine Candidate Adjuvanted with Porcine Ferritin Scaffold for African Swine Fever Virus

**DOI:** 10.3390/vaccines13060585

**Published:** 2025-05-30

**Authors:** Lidan Sun, Yuping Ding, Jingqi Niu, Yingjun Li, Zeliang Chen

**Affiliations:** 1Key Laboratory of Livestock Infectious Diseases in Northeast China, Ministry of Education, College of Animal Science and Veterinary Medicine, Shenyang Agricultural University, Shenyang 110866, China; sunlidan0399@163.com (L.S.); njq2022200177@163.com (J.N.); 2Twins Group Co., Ltd., Nanchang 330046, China; 19907126687@163.com; 3Beijing Tonghe Litai Biotechnology Co., Ltd., Beijing 102600, China

**Keywords:** African swine fever (ASF), Nanovaccine, T-cell epitopes, porcine ferritin (pFTH1), IFN-γ^+^

## Abstract

**Background**: African swine fever (ASF) is a highly contagious acute febrile disease with a near 100% mortality rate. There are currently no safe and effective vaccines for this disease. Cellular immunity plays an important role in the process of anti-viral, activating an effective cellular immune response is a prerequisite for the effectiveness of the vaccine. **Methods**: To effectively activate cellular immune responses, 133 immunodominant T cell epitopes (TEPs) were identified and synthesized into ten recombinant multi-epitope proteins (MEPs). These MEPs were subsequently conjugated to porcine ferritin (pFTH1) to generate MEPs-pFTH1 nanoparticles. Animal experiments were conducted to evaluate their immunogenicity and biocompatibility. **Results**: Animal experiments demonstrated that both MEPs and MEPs-pFTH1 nanoparticles induced significant humoral and cellular immune responses. Compared to MEPs monomers, the MEPs-pFTH1 nanoparticles induced a 10- to 100-fold increase in IgG and IgG2a antibody titers (*p* < 0.05), as well as a significantly higher number of IFN-γ^+^ cells. Serum from pigs immunized with MEPs-pFTH1 nanoparticles can significantly inhibit ASFV replication. **Conclusions**: Our novel self-assembled porcine ferritin nanovaccine candidate can induce strong humoral and cellular immune responses in swine and mice that effectively inhibit ASFV replication. Therefore, the nanovaccine is a highly biocompatible and safe candidate vaccine for ASF that warrants further investigation, such as conducting animal challenge experiments to evaluate the effectiveness of the vaccine.

## 1. Introduction

African swine fever (ASF) is an acute, febrile, and highly lethal infectious disease in swine, caused by the African swine fever virus (ASFV). The mortality rate among swine infected with a highly pathogenic ASFV strain can reach as high as 100% [1,2]. It is widely recognized that vaccination is the most effective and economical approach to controlling the spread of infectious diseases. Currently, there are no commercially available vaccines or drugs for ASF. Traditional inactivated vaccines, subunit vaccines, and DNA vaccines are considered safe but fail to provide complete and long-lasting protection [3,4,5,6,7,8].

Current research suggests that attenuated vaccines can offer varying degrees of protection; however, they are associated with residual toxicity, potential safety risks, and an inability to fully eliminate the virus [9,10,11]. Therefore, new antigen delivery and adjuvant strategies are essential for the development of safer and more effective vaccines.

Epitope vaccines, as a novel class of vaccines, have garnered significant attention in the realms of tumor, viral, and bacterial vaccine research. Unlike traditional vaccines, which may not encompass all amino acid regions capable of eliciting an effective immune response, epitope vaccines are designed to enhance immune activation. During the immune response, various factors, including antigen uptake and processing, the antibodies generated by non-suppressive epitopes, and the negative regulation imposed by immunosuppressive epitopes, can severely deplete the body’s limited immune potential. Epitope vaccines offer several advantages, including strong specificity, safety, stability, and ease of purification and expression, positioning them as a promising avenue for the development of next-generation subunit vaccines [12,13]. Furthermore, bacterial proteins with self-assembling properties provide several benefits for antigen delivery, such as high biocompatibility, stability, symmetry, and multivalency [14]. The size of the particles is a crucial parameter influencing the pharmacokinetics of vaccines; nanoparticles smaller than 5 nm can readily enter systemic circulation, while those ranging from 10 to 100 nm tend to accumulate in lymph nodes through lymphatic circulation [15,16,17,18,19,20,21].

Like viral particles, self-assembling nanoparticles can display antigens on their surfaces at a high density [22]. In comparison to soluble proteins, multivalent antigens are more effective at inducing B cell receptor (BCR) cross-linking, which significantly enhances the uptake and presentation of both high-affinity and low-affinity antigens. While soluble proteins are predominantly internalized by B cells with high-affinity BCRs, low-affinity B cells may also play a role in the production of neutralizing antibodies (NAbs) [14,23]. Research shows that multivalent antigens can enhance T cell activation and increase the production of specific antibodies by B cells up to tenfold [24,25,26]. Consequently, nanoparticles can elicit stronger humoral and cellular immune responses at lower doses [27,28]. In the past decade, bacterial protein-based nanoparticles have been evaluated in numerous subunit vaccine formulations [29,30,31,32,33], especially ferritin [34,35,36,37,38,39]. Ferritin nanoparticles are hollow cages composed of 24 identical subunits, with inner and outer diameters of 8 and 12 nm, respectively [22,40]. These antigen delivery vectors exhibit greater stability and multivalency compared to traditional subunit vaccines, and they have been widely used in the formulation of various vaccines. It has been shown that ferritin-based nanovaccines are 10–50 nm in diameter, which is an ideal size for interacting with different immune cells and inducing effective immune responses [22].

ASFV is a large DNA virus comprising over 150 proteins, the majority of which remain functionally uncharacterized [41]. Previous studies of ASFV subunit vaccines demonstrated that one or several target ASFV proteins were inadequate to confer full protection [5]. In our previous work, 15 immunodominant T cell epitopes (TEPs) from ASFV were identified using the immunoinformatics prediction method and synthesized into recombinant multi-epitope proteins (TEP), which were subsequently conjugated to AaLS scaffolds to generate TEP-Spy-AaLS nanoparticles. We found that these TEP-Spy-AaLS nanoparticles elicited stronger humoral and cellular immune responses in both mice and pigs compared to TEP monomers [42]. Based on this, the present study identified 133 immunodominant TEPs within ASFV proteins that are conserved across 12 prevalent strains reported in the past decade. These TEPs were synthesized into ten different recombinant MEPs, each of which was then conjugated to porcine ferritin (pFTH1) via the SpyCatcher/SpyTag system to create MEPs-pFTH1 nanoparticles. Our animal experiments demonstrated that while MEPs induced significant cellular and humoral immune responses, the MEPs-pFTH1 nanoparticles elicited more robust immune responses, and effectively inhibited ASFV replication. This suggests their potential as a promising candidate vaccine for ASF.

## 2. Materials and Methods

### 2.1. Prediction of ASFV TEPs and Screening of Immunodominant TEPs

The amino acid sequences of 12 prevalent ASFV strains in the past decade “https://www.ncbi.nlm.nih.gov/ (accessed on 1 September 2022)” were compared using the MAFFT algorithm in SnapGene to identify highly conserved ASFV proteins (>90% sequence homology). Linear MHC-I-restricted TEPs in each conserved ASFV protein were predicted using the Immune Epitope Database“IEDB: http://www.iedb.org/ (accessed on 1 September 2022)”, and their binding affinity was evaluated using NetMHCpanEL 4.1 [43,44]. One to two immunodominant TEPs (with the lowest %Rank scores) were selected from each conserved ASFV protein and cross-referenced with B cell epitopes to identify overlapping epitopes. Detailed information on the 12 ASFV strains and highly conserved proteins are shown in Appendix A. Dominant TEP amino acid sequences and immunogenicity scores are summarized in Appendix A.

### 2.2. Construction of Recombinant MEPs

Ten recombinant MEPs (MEP1 to MEP10) were constructed by linking 10–15 TEPs into single protein constructs, each with a molecular weight of approximately 20–25 kDa. The specific substrate sequence PMGAP of cathepsin S (Cat) was inserted between the TEPs in each MEP construct to facilitate their release following phagocytosis by immune cells. The SpyTag (AHIVMVDAYKPTK) and 6× His tags were respectively inserted into the N-terminus and C-terminus of the MEP via a flexible linker peptide (GSSS)_3_, while restriction sites (NdeI, XhoI) were introduced at both ends. After optimizing the nucleotide codon sequence, the sequences were sent to GeneRay for synthesis. Then the target gene fragments were ligated to pET-28a vectors to construct pET-28a-SpT-MEP plasmids, which were then chemically transformed into *Escherichia coli* BL21 competent cells for protein expression.

### 2.3. Construction of pFTH1 Plasmids

Porcine ferritin (pFTH1) was used in this study to reduce the generation of antibodies against heterologous ferritin, which could lead to premature clearance of the antigen-presenting ferritins and decrease the vaccine’s efficacy. The SpyCatcher (113aa) and 6× His tags were respectively inserted at the N- and C-termini of the heavy chain of pFTH1 (GenBank: XP_005660860.1) via a flexible linker peptide (GSSS)_3_, while restriction sites (NdeI, XhoI) were introduced at both ends. After codon sequence optimization, the sequence was sent to BGI Genomics for synthesis. The target gene fragment was then ligated to the pET-28a vector to create a pET-28a-SpC-pFTH1 plasmid, which was subsequently chemically transformed into *E. coli* BL21 competent cells for protein expression.

### 2.4. Expression and Purification of Recombinant MEPs and pFTH1 Proteins

*E. coli* BL21 cells that stably express the MEPs and pFTH1 were inoculated into 1 L of fresh LB medium (1:50) and cultured at 37 °C with 220 revolutions per minute (Rotor diameter 20 cm), shaking until OD600 0.4–0.6. The bacterial cultures were supplemented with 0.5 mM IPTG and incubated for another 16–18 h at 16 °C with 200 revolutions per minute (Rotor diameter 20 cm) shaking to induce protein expression. Bacteria were collected and lysed, and the target proteins were purified via an HIS-select nickel affinity gel column (Sigma Aldrich, St. Louis, MO, USA, Cat. P6611) according to the manufacturer’s instructions. The inclusion bodies were first denatured and solubilized in 8 M urea prior to protein purification and subsequently refolded into their active conformation by gradually reducing the urea concentration (8 M, 6 M, 4 M, 2 M, 1 M, 0 M) in the renaturation buffer (50 mM Tris-HCl pH 8.0, 150 mM NaCl, 300 mM arginine, 5% (*v*/*v*) glycerol, 0.5 mM GSH and 2 mM GSSG). Each concentration gradient required at least 6 h of dialysis, which was carried out at 4 °C once the urea concentration was reduced to 4 M. The purified proteins were desalted and concentrated using ultrafiltration tubes, and the concentrations of the target proteins were determined using a BCA protein assay kit (Thermo Fisher, Waltham, MA, USA).

### 2.5. Preparation of MEPs-pFTH1 Nanoparticles

To explore the optimal formulation, the MEPs with SpyTag and pFTH1 proteins with SpyCatcher were mixed and assembled in TBS buffer at various ratios (1.5:1, 2:1) at 4 °C overnight. During this process, the MEPs covalently bind to the pFTH1 scaffold via the SpyTag/SpyCatcher interaction, resulting in the formation of ferritin self-assembled nanoparticles that display MEPs on the surface (Figure 1). The conjugated samples were analyzed using SDS-PAGE to identify the optimal assembly ratio, and subsequent MEPs-pFTH1 nanoparticles were assembled according to the optimal ratio. The nanoparticles were purified through exclusion chromatography and agarose gel microsphere NW Rose 6FF (Nano Micro) with a flow rate of 0.5 mL/min, and then stored in phosphate buffer solution (pH7.4), at −80 °C for future use.

### 2.6. Western Blot

The antigenicity of the synthesized MEPs and MEP-pFTH1 nanoparticles was evaluated by Western blot (WB). Briefly, samples were mixed with an appropriate volume of 5 × SDS loading buffer (containing β-ME, DTT) and heated in boiling water for 10 min. The proteins (2 µg per lane) were separated by 12% SDS-PAGE and then transferred to a nitrocellulose (NC) membrane using an electrophoretic transfer device. The NC membrane was blocked in PBS-T containing 5% skimmed milk powder for 1 h at room temperature, incubated with PBS-T containing 1:1000 ASFV positive serum (collected from an ASFV-infected pig farm) overnight at 4 °C, washed 3 times in PBS-T (5 min/wash), and incubated with PBS-T containing 1:2000 HRP-labeled goat anti-pig IgG antibody (Solarbio) for 1 h at room temperature. The NC membrane was washed three times, incubated with enhanced chemiluminescence (ECL) substrate solution (Tanon, Shanghai, China), and visualized using the film-based autoradiography.

### 2.7. Transmission Electron Microscopy (TEM)

MEPs-pFTH1 nanoparticles were adjusted to a concentration of 0.2 mg/mL and applied to the surface of a 300-mesh copper grid. After a 3-min incubation, the excess sample was removed using absorbent paper, and the copper grid was negatively stained with fresh 2% (*w*/*v*) uranyl acetate for 45 s, followed by 5 to 10 min of drying at room temperature. The morphology of the nanoparticles was observed and photographed at 100 K magnification using a transmission electron microscope (Hitachi HT7800, Hitachi High-Tech, Tokyo, Japan).

### 2.8. Immunization of Mice and Sample Collection

Twenty-four 6- to 8-week-old SPF BALB/c female mice were purchased from Beijing SPF Biotechnology Co., Ltd. and housed in a barrier facility in accordance with the “Guide for the Breeding and Use of Laboratory Animals”. All experimental procedures were approved by the Experimental Animal Ethics Committee of Beijing Tonghe Litai Biotechnology Co., Ltd. (approval number: 2023-072). Mice were randomly divided into the MEPs monomer group, the MEPs-pFTH1 nanoparticle group, and the negative control group (PBS), with four mice per group. The immunological substances were mixed with an equal volume of 206 adjuvant (Seppic, La Garenne Colombes, France) and administered via multi-site subcutaneous injection at a dose of 0.5 nM (approximately 12.25 µg) of antigen. Three immunizations were carried out at 2-week intervals, and blood was collected from the retro-orbital venous plexus two weeks after each immunization and three weeks after the third immunization. Serum was collected for the detection of IgG, IgG2a, and IgE antibodies by ELISA. At two weeks after the third immunization, four mice from each group were euthanized by intraperitoneal injection of 50 µg/g pentobarbital sodium, and the spleen was aseptically harvested for the isolation of splenic lymphocytes using the Mouse Splenic Lymphocyte Isolation Kit (Solarbio, Cat. P860) for the ELISpot assay.

### 2.9. Immunization of Pigs and Sample Collection

Fifteen adult Bama miniature pigs (25 ± 1 kg) were provided by Beijing Tonghe Litai Biotechnology Co., Ltd., Beijing, China. and housed in a barrier facility in accordance with the “Guide for the Breeding and Use of Laboratory Animals” of this company. All experimental procedures were approved by the Experimental Animal Ethics Committee of Beijing Tonghe Litai Biotechnology Co., Ltd. (approval number: 2023-056). The pigs were randomly divided into five groups: the MEPs monomer group, MEPs-pFTH1 nanoparticle group, pFTH1 only group, AaLS only group, and blank control group (PBS), with three pigs per group. The immune substances were mixed with an equal volume of 206 adjuvant and administered via intramuscular injection at a dose of 10 nM (approximately 245 µg) of antigen. The second and third doses were administered at 2-week intervals following the first dose. Blood was collected from the anterior vena cava on Days 7 and 14 after each dose and in the third week after the third immunization. Serum was separated by centrifugation at 4 °C and 2000× *g* for 10 min, then stored at −20 °C for future use. Anticoagulated blood was collected, and PBMCs were aseptically isolated using the Pig Peripheral Blood Mononuclear Cell Isolation Kit (Solarbio, Beijing, China) for the ELISpot assay.

### 2.10. Enzyme-Linked Immunosorbent Assay

Purified TEP-pFTH1 nanoparticles were diluted to a concentration of 1 µg/mL in coating buffer (0.05 mol/L carbonate buffer, pH 9.6). A volume of 100 µL was added at 100 µL/well of a 96-well microplate (Corning, Corning, NY, USA) and incubated overnight at 4 °C. The microplate was washed 5 times in PBS-T, blocked with 100 µL/well of PBST containing 5% skimmed milk powder at 37 °C for 1 h, incubated with 100 µL/well of various dilutions (1:10^1^, 10^2^, 10^3^, 10^4^, 10^5^, 10^6^, 10^7^, 10^8^) of serum at 37 °C for 1 h, washed 5 times, and then incubated with 100 µL/well of 1:5000 HRP-labeled goat anti-mouse IgG antibody (ZSGB-BIO), 1:2000 HRP-goat anti-piglet IgG antibody (Solarbio, China), 1:10,000 HRP-goat anti-mouse IgG2a antibody (Bethyl), or 1:10,000 HRP-goat anti-mouse IgE antibody (Gene Tex, Irvine, CA, USA) at 37 °C for 45 min. The plate was washed 5 times and incubated with 100 µL/well of TMB substrate solution at room temperature for 15 min in the dark, followed by the addition of an equal volume of stop solution (H_2_SO_4_) into each well. Absorbance was measured at 450 nm using the SpectraMax Plus microplate reader (Thermo Fisher, USA).

### 2.11. Enzyme-Linked Immunospot Assay

Single-cell suspensions were prepared using the Mouse Spleen Lymphocytes Kit (Solarbio, product number: P860) and the Porcine PBMCs Isolation Kit (Solarbio, product number: P420), and were adjusted to a density of 5 × 10^6^ cells/mL. ELISpot assays were conducted according to the instructions provided with the ELISpot Kit (Mabtech, Nacka Strand, Sweden). Briefly, the microplate was pretreated, rehydrated, and blocked with 200 µL/well of RPMI-1640 supplemented with 10% fetal bovine serum for 30–60 min. After discarding the blocking solution, 100 μL of medium, 50 μL of 1 μg/mL MEP monomer stimulator, and 50 μL of the cell suspension were sequentially added to each well and incubated at 37 °C with 5% CO_2_ for 18–48 h. The plate was then washed 5 times with 200 μL/well of PBS, followed by incubation with 100 μL/well of 0.5 μg/mL detection antibody (in PBS containing 0.5% FBS) at room temperature for 2 h. Subsequently, the plate was washed 5 times with PBS-T and incubated with 100 μL/well of 1:000 streptavidin ALP solution (in PBS containing 0.5% FBS) at room temperature for 2 h. After washing, the plate was incubated with 100 μL/well of BCIP/NBT in the dark until colored spots were observed (generally not exceeding 30 min). The plate was rinsed with tap water to terminate color development and then dried in the dark. Spots were counted using the Immunospot^®^ S5 analyzer within 7 days of assay completion.

### 2.12. Serum Virus Neutralization Assay

Porcine alveolar macrophages (PAMs) derived from healthy piglets, isolated and cryopreserved by our research team in the previous work), were seeded at a density of 1 × 10^5^ cells per well in 96-well microplates pre-coated with poly-lysine (Phygene, Fuzhou, China) and incubated for 24 h. A half-volume medium change was performed the following day, and the cells were further cultured for an additional 24 h. Inactivated sera from pigs injected with MEPs-pFTH1 nanoparticles, sera from naturally infected and recovered pigs, and sera from healthy pigs were mixed with 0.1 MOI of ASFV at different ratios and incubated at 37 °C for 1 h (three replicates for each dilution). After removing the medium from the microplates, the serum-virus mixtures were added to the 96-well microplates and incubated at 37 °C for 2 h, after which they were discarded. RPMI-1640 medium containing 3% pig serum (Feimobio, Beijing, China) was then added to the microplates and cultured at 37 °C with 5% CO_2_ for another 96 h. The virus was released from the cells into the medium by subjecting the microplates to three cycles of repeated freezing and thawing. The medium was collected and centrifuged at 1500× *g* for 10 min to remove cell debris.

### 2.13. Quantitative PCR

Viral DNA was extracted from the samples using a viral DNA purification kit (TaKaRa, Kusatsu, Japan). The viral genome copy number, indicated by the B646L gene, was quantified using qPCR (Bio-Rad Laboratories, Inc., Hercules, CA, USA) in a 10 µL reaction containing 1 µL genomic DNA, 0.4 µL (10 µmol) forward primer (5′-ATAGAGATACAGCTCTTCCAG-3′), 0.4 µL (10 µmol) reverse primer (5′-GTATGTAAGAGCTGCAGAAC-3′), 5 µL 2× ChamQ Universal SYBR qPCR Master Mix (Vazyme, Nanjing China), and 3.2 µL sterile water. The qPCR conditions were 95 °C for 30 s, followed by 40 cycles of 95 °C for 10 s and 60 °C for 30 s. Melting curve analysis was performed using the instrument’s default program. Viral genome copy numbers were calculated from a standard curve based on B646L gene amplification.

### 2.14. Statistical Analysis

Data are presented as mean ± standard error of the mean from at least three replicates. All data sets were analyzed using one- or two-way ANOVA considering a normal (Gaussian) distribution followed by a Tukey test for multiple comparisons to identify specific differences between treatment groups. The statistical analysis was performed using Prism software 10.4.0 (GraphPad, Boston, MA, USA). The abbreviation “ns” indicates no significant difference (*p* > 0.05); * indicates a significant difference (*p* < 0.05); ** indicates a highly significant difference (*p* < 0.01); and *** indicates an extremely significant difference (*p* < 0.001).

## 3. Results

### 3.1. Screening of Immunodominant TEPs and Construction of MEPs-pFTH1 Particles

The process of T cell epitope prediction and the construction of the nanoparticle vaccine is shown in Figure 1. Bioinformatic analysis of the amino acid sequences of 12 prevalent ASFV strains revealed 105 relatively conserved proteins with over 90% amino acid sequence homology and 170,568 TEPs (Appendix A). From each conserved protein, one to two immunodominant TEPs were selected, resulting in a total of 133 TEPs. Figure 2A illustrates the positions of these 133 TEPs within the ASFV genome. The selected TEPs covered 84.76% of the conserved ASFV proteins, which comprised 13.48% structural proteins and 86.52% non-structural proteins (Figure 2B,C). A total of 10 different recombinant MEPs (MEP1 to MEP10) from the 133 TEPs were generated by linking 10–15 TEPs into each MEP construct. The number of structural and non-structural proteins in each MEP is shown in Figure 2D. Furthermore, the amino acid sequences of the selected TEPs partially or completely overlapped with those of B cell epitopes, indicating their potential to induce humoral immune responses (Appendix A).

### 3.2. Characterization of MEPs-pFTH1 Nanoparticles

The recombinant MEP and pFTH1 plasmids were transformed into *E. coli* and the expression form of the target protein were analyzed. The results showed that MEP6, MEP8, MEP9, MEP10, and pFTH1 scaffold protein were expressed in soluble form, whereas MEP1, MEP2, MEP3, MEP4, MEP5, and MEP7 were expressed as inclusion bodies. SDS-PAGE analysis of the purified target proteins revealed band sizes consistent with the theoretical molecular weight, with only minimal impurity bands detected. This result indicates that all 10 MEPs were successfully expressed and purified (Figure 3A).

Next, the MEPs were conjugated to pFTH1 proteins at various ratios to determine the optimal ratio for nanoparticle construction. SDS-PAGE and Coomassie brilliant blue staining showed that a 1:1 assembly ratio resulted in the lowest quantity of free MEP monomers and scaffolds, achieving a conjugation efficiency of over 90% (Figure 3B). WB analysis demonstrated that the MEPs-pFTH1 nanoparticles were recognized by specific antibodies in ASFV-positive sera, appearing as a single band at the expected molecular weight (Figure 3C). This confirms that all 10 MEPs exhibit ASFV antigenicity. Furthermore, the MEPs-pFTH1 nanoparticles displayed a dispersed and uniformly sized granular morphology under TEM, with an average particle size of approximately 16 nm (Figure 3D).

### 3.3. pFTH1 Is Minimally Immunogenic in Pigs

To assess the immunogenicity of the scaffold protein, pigs were injected with either empty pFTH1 scaffold proteins or AaLS (*Aquifex aeolicus* lumazine synthase) scaffold proteins. Blood samples were subsequently collected for the detection of specific antibodies and IFN-γ+ cells by ELISA and ELISpot assays, respectively. The results showed that pFTH1 induced significantly lower titers of anti-scaffold antibodies and a reduced number of IFN-γ+ cells compared to AaLS (Figure 4). These findings suggest that pFTH1 exhibits greater biocompatibility and reduced immunogenicity, potentially minimizing the risk of premature clearance of the nanoparticles by anti-scaffold antibodies.

### 3.4. MEPs-pFTH1 Nanoparticles Are Highly Biocompatible

Hematology and serum biochemistry parameters serve as important indicators of health status. The safety of the nanoparticles was evaluated by measuring these parameters in immunized pigs. It was observed that all hematological parameters remained within their normal reference ranges for pigs that received the MEPs monomer and MEPs-pFTH1 nanoparticles. Although the MEPs monomer and MEPs-pFTH1 groups exhibited increased percentages of lymphocytes and white blood cells compared to the control group, these values were still within the normal reference range and comparable between the two groups (*p* > 0.05) (Table 1). Such changes were likely attributed to the immune responses elicited by the antigens. Furthermore, no significant differences in serum biochemistry parameters were found among the MEPs monomer, MEPs-pFTH1, and control groups (*p* > 0.05) (Table 2), suggesting that the nanoparticles demonstrated high biocompatibility and safety in pigs.

### 3.5. MEPs-pFTH1 Nanoparticles Induce Strong Humoral and Cellular Immune Responses in Mice

To evaluate the immunogenicity of the test antigens, we injected mice with three doses of either the MEPs-pFTH1 nanoparticles or MEPs monomers, with a 2-week interval between each dose (Figure 5A). ELISA showed that specific IgG titers in the serum were generally comparable between the MEPs-pFTH1 and MEPs monomer groups in the second week following the first dose and second dose (Figure 5B,C). However, the serum titers of IgG specific for MEP1, MEP4, MEP7, MEP8, and MEP9 were significantly higher in MEPs-pFTH1 groups in the second week following the third dose (Figure 5D), as well as higher serum titers of IgG-specific for serum titers of IgG specific for all MEPs-pFTH1 group (except MEP3 and MEP5) at the third week following the third dose (Figure 5E), compared to the MEPs monomer group (all *p* < 0.05). Similarly, serum titers ofIgG2a specific for all MEPs (except MEP5 and MEP7) were significantly higher in the MEPs-pFTH1 group than in the MEPs monomer group than in the MEPs monomer group at the second week following the third dose (all *p* < 0.05). In contrast, there were no significant differences in serum-specific IgE titers between the MEPs-pFTH1 group and the MEP monomer group (Figure 5G). Collectively, these data demonstrate that MEPs-pFTH1 nanoparticles elicited more potent immune responses compared to the MEPs monomers. IgG2a is a signature antibody subclass of the Th1-type immune response. An increase in IgG2a levels indicates activation of Th1 cells-, which may reflect a stronger cellular immune response and enhanced protective effect.

Furthermore, we examined the number of IFN-γ+ cells induced by MEPs using ELISpot. The experimental data showed that both MEPs monomers and MEPs-pFTH1 significantly increased the number of IFN-γ+ splenic lymphocytes compared to the control in mice. Moreover, the number of IFN-γ+ splenic lymphocytes was significantly higher in all MEPs-pFTH1 groups (except for MEP3-pFTH1) than in their respective MEPs monomer groups (all *p* < 0.05) (Figure 5H). Taken together, the Th1-type response is mainly driven by IFN-γ, these findings indicate that MEP-pFTH1 nanoparticles induce stronger cellular immune responses in mice compared to the MEP monomers.

### 3.6. MEPs-pFTH1 Nanoparticles Induce Robust Humoral and Cellular Immune Responses in Pigs

Given that pigs are the natural hosts of ASFV, pigs were injected with the MEPs antigens to evaluate the potential safety and efficacy of nanoparticles (Figure 6A). Consistent with the trends observed in the mouse study, serum IgG titers in pigs were elevated compared to the control group in the second week following the first and second doses, with levels comparable between the MEPs-pFTH1 and MEPs monomer groups (Figure 6B,C). In the second and third week after the third dose, serum IgG titers were significantly higher (10- to 100-fold) in all MEPs-pFTH1 groups compared to their respective MEPs monomer groups (Figure 6D,E). Similarly, the number of IFN-γ+ PBMCs increased after the injection of both MEPs monomers and MEPs-pFTH1 nanoparticles, with levels significantly higher in the MEPs-pFTH1 groups than in their respective MEPs monomer groups after the second and third doses (Figure 6F,I). Collectively, these findings further confirm that the MEPs-pFTH1 nanoparticles can induce robust humoral and cellular immune responses in pigs, surpassing those elicited by MEPs monomers alone.

### 3.7. Immune Serum from Pigs Injected with the MEPs-pFTH1 Nanoparticles Significantly Inhibits ASFV Replication

The animal experiments demonstrated that the MEPs-pFTH1 nanoparticles significantly increased serum titers of IgG and IgG2a, as well as the number of IFN-γ+ cells. Based on these findings, we conducted a serum virus neutralization assay to assess the impact of these immune molecules on ASFV replication. We observed significant morphological changes in PAMs treated with ASFV + Control, characterized by increased cell diameter, cell aggregation, and cellular disruption. However, these changes in PAMs were reduced in the ASFV + MEPs-pFTH1 immune serum group and the ASFV + convalescent serum group. Subsequently, we quantified the viral genome copy number by qPCR to evaluate the inhibitory effect of the immune serum on ASFV proliferation. Our data showed that a 1:16 dilution of both MEPs-pFTH1 immune serum and convalescent serum markedly inhibited ASFV replication, with no significant observed difference between the two sera (Figure 7). In conclusion, these experimental data confirm that the immune mediators induced by MEPs-pFTH1 nanoparticles effectively inhibit ASFV replication, supporting the potential application of this nanovaccine in the prevention and control of ASFV.

## 4. Discussion

Vaccination has constituted the strategy of choice to prevent the propagation of infectious diseases [45]. In comparison to inactivated or live-attenuated pathogen-based vaccines, subunit vaccines constitute safer formulations, although they tend to be weakly immunogenic and require the use of non-specific immunostimulants [46]. Over the last two decades, the interest in using self-assembling peptides and proteins to conceive supramolecular assemblies for nanovaccine design has increased considerably [47]. Ferritin is a self-assembly antigen presentation platform recognized for its exceptional thermal stability, pH stability, biocompatibility, biodegradability, and cost-effectiveness, rendering it ideal for large-scale vaccine production [48]. The distribution of antigens is crucial for vaccine efficacy. The multivalent presentation of antigens on its surface enhances their uptake by dendritic cells and macrophages, facilitating subsequent drainage to lymph nodes and thereby improving vaccine effectiveness [49]. Currently, vaccines developed based on ferritin-conjugated antigens for multiple pathogens have entered the clinical evaluation stage, such as vaccines based on ferritin for human immunodeficiency virus type 1 (NCT05903339), influenza A (NCT03186781, NCT03814720, NCT04579250, and NCT05155319), Epstein-Barr virus (NCT04645147 and NCT05683834), and SARS-CoV-2 (NCT04784767 and NCT06147063) are currently in clinical trials. Phase 1 clinical trials of multiple influenza A ferritin vaccines (NCT03186781 and NCT03814720) have shown that these vaccines are safe and well-tolerated. The safety of ferritin as a vaccine platform has been confirmed, making it an ideal system for developing effective vaccines. In this study, to further optimize the biocompatibility of the nanovaccine and minimize the premature clearance of antigen-scaffold complexes, we employed porcine ferritin as the antigen delivery platform to mitigate the generation of anti-scaffold antibodies. Cellular immunity is critical for the development of vaccine-induced long-term immunological memory and protection. Effector T cells that secrete Th1 cytokines such as IFN-γ play an important role in activating macrophages, neutrophils, and CD8+ T cells during viral infections. Therefore, quantifying IFN-γ+ cells following antigen stimulation via ELISpot provides a valuable approach for assessing vaccine effectiveness.

Traditional vaccines do not induce an effective immune response in all amino acid regions. During an immune response, the uptake and processing of antigens, the production of antibodies by non-suppressive epitopes, and negative immune regulation by immunosuppressive epitopes all contribute to the depletion of the body’s limited immune capacity. In contrast, epitope vaccines are highly specific, safe, and stable, and are easily purified and expressed, representing a promising direction for the development of next-generation subunit vaccines [12,13]. Furthermore, epitope vaccines are more effective at eliciting immune responses against mutant virus strains than traditional vaccines. The selection of antigenic epitopes is critical for determining the immunogenicity of epitope vaccines; thus, an ideal vaccine should encompass as many immunodominant epitopes as possible. This study systematically predicted the dominant linear MHC class I epitopes in all conserved proteins of ASFV using NetMHCpanEL 4.1 and successfully identified 133 dominant T EPs. Ten recombinant MEPs were constructed and expressed, all of which exhibited the antigenic characteristics of ASFV. We conjugated them to pFTH1 via the SpyTag/SpyCatcher system, resulting in the formation of MEPs-pFTH1 self-assembling ferritin nanoparticles. Our animal experiments indicated that these nanoparticles did not adversely affect the blood parameters of immunized animals, demonstrating high biocompatibility and a strong safety profile. Moreover, MEPs-pFTH1 induced IgG and IgG2a titers that were 10–100 times higher, along with 2–3 times more IFN-γ+ cells in animals compared to MEP monomers alone, suggesting that the nanoparticles triggered stronger humoral cellular immune responses. Previous studies utilizing ferritin self-assembled nanoparticle antigens, constructed through immunoimmune informatics tools to predict T and B cell epitopes of ASFV target antigens, successfully induced robust humoral and cellular immune responses. However, none of these studies conducted an animal challenge test [42,50]. Compared to traditional attenuated vaccines, epitope vaccines offer enhanced safety and have demonstrated favorable outcomes in numerous studies [45,46,47,48,49,50,51,52].

Although nanoparticle vaccines have successfully elicited robust humoral and cellular immune responses in the mouse and pig models, this study still has several limitations that deserve further discussion. Firstly, the detection of immune markers in pigs was limited by the relatively small number of available reagents and kits, as well as the difficulty in procurement. This has, to some extent, restricted the ability to conduct a comprehensive evaluation of the immune effects of nanoparticle antigens. Secondly, the isolation of mouse splenic lymphocytes was required for the ELISpot assay. To ensure the acquisition of highly viable cell samples, we had to limit the experimental sample size to a small scale (N = 4). Thirdly, due to the need for an Animal Biosafety Level 3 (ABSL-3) facility, we were unable to perform a virus challenge test to assess the protective effect of the vaccine. However, we demonstrated the potential protective effect of the vaccine by showing that sera from pigs vaccinated with the nanoparticle vaccine significantly inhibited ASFV replication in vitro in the serum micro-neutralization assay.

## 5. Conclusions

We have successfully developed a novel MEPs-pFTH1 self-assembling candidate nanovaccine that induces strong humoral and cellular immune responses in both pigs and mice. Additionally, immune sera from vaccinated animals significantly inhibited ASFV replication in vitro. Therefore, the MEPs-pFTH1 nanoparticles are promising ASF vaccine candidates with high biocompatibility and safety worthy of further research, such as conducting animal challenge experiments to evaluate the effectiveness of the vaccine.

## Figures and Tables

**Figure 1 vaccines-13-00585-f001:**
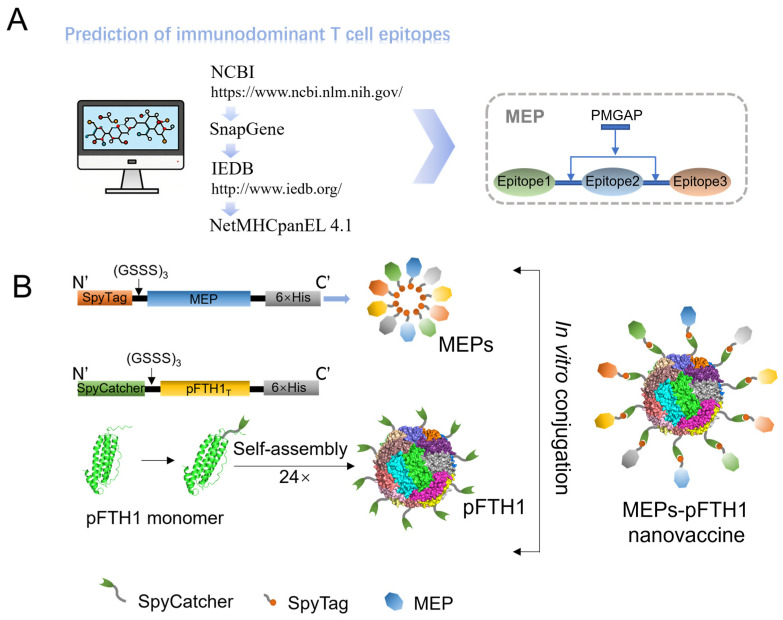
Schematic diagram of MEPs-pFTH1 nanoparticles. (**A**) Schematic diagram of a MEP construct comprising several TEPs in tandem. (**B**) Schematic diagram of MEPs-pFTH1 construction.

**Figure 2 vaccines-13-00585-f002:**
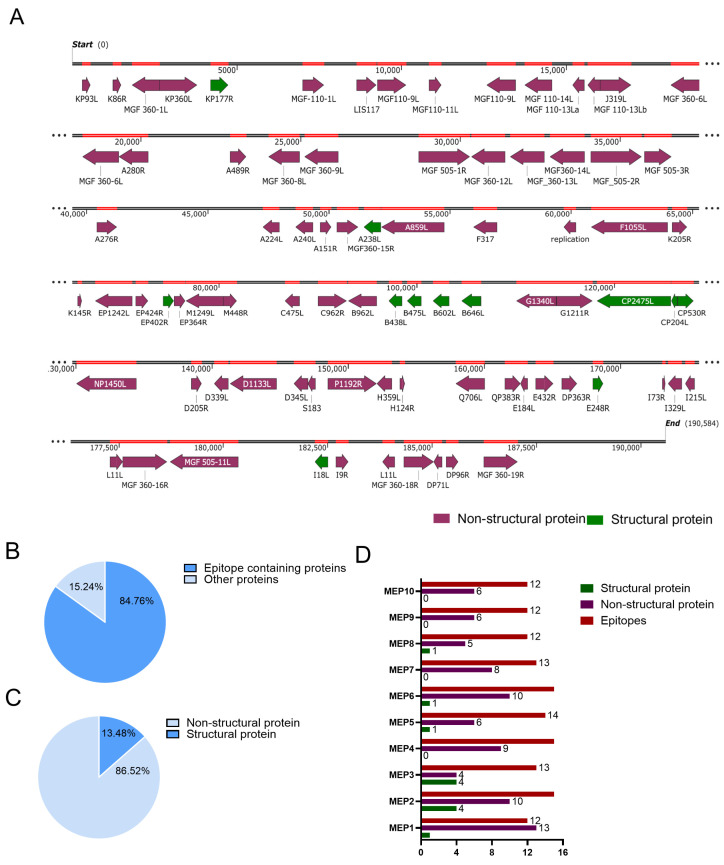
Screening of immunodominant TEPs in conserved ASFV proteins. (**A**) The positions of the TEPs in the ASFV genome. The purple line represents non-structural proteins, and the green line represents structural proteins. (**B**) Epitope coverage across the conserved ASFV proteins. (**C**) The percentage of structural and non-structural proteins among the conserved ASFV proteins. (**D**) The number of structural proteins, non-structural proteins, and TEPs contained in each MEP.

**Figure 3 vaccines-13-00585-f003:**
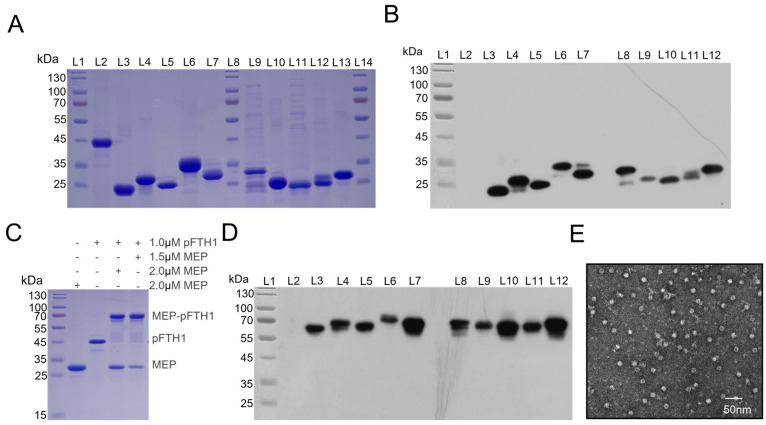
Morphology and antigenicity of MEPs-pFTH1 nanoparticles. (**A**) Coomassie staining of purified MEPs by SDS-PAGE, L1: protein ladder, L2: pFTH1 (39 kDa), L3: MEP1 (20 kDa), L4: MEP2 (25 kDa), L5: MEP3 (22 kDa), L6: MEP4 (30 kDa), L7: MEP5 (27 kDa), L8: protein ladder, L9: MEP6 (27 kDa), L10: MEP7 (23 kDa), L11: MEP8 (21 kDa), L12: MEP9 (22 kDa), L13: MEP10 (26 kDa), L14: protein ladder. (**B**) WB of MEPs using ASFV-positive serum, L1: protein ladder, L2: pFTH1 (39 kDa), L3: MEP1 (20 kDa), L4: MEP2 (25 kDa), L5: MEP3 (22 kDa), L6: MEP4 (30 kDa), L7: MEP5 (27 kDa), L8: MEP6 (27 kDa), L9: MEP7 (23 kDa), L10: MEP8 (21 kDa), L11: MEP9 (22 kDa), L12: MEP10 (26 kDa), (**C**) Coomassie staining of MEP-pFTH1 nanoparticles assembled at various ratios by SDS-PAGE. (**D**) WB of MEP-pFTH1 nanoparticles using ASFV-positive serum, L1: protein ladder, L2: pFTH1 (39 kDa), L3: MEP1-pFTH1(59 kDa), L4: MEP2-pFTH1 (64 kDa), L5: MEP3-pFTH1 (61 kDa), L6: MEP4-pFTH1 (69 kDa), L7: MEP5-pFTH1 (66 kDa), L8: MEP6-pFTH1 (66 kDa), L9: MEP7-pFTH1 (62 kDa), L10: MEP8-pFTH1 (60 kDa), L11: MEP9-pFTH1 (61 kDa), L12: MEP10-pFTH1 (65 kDa). (**E**) Morphology of MEPs-pFTH1 nanoparticles under TEM (×100 K).

**Figure 4 vaccines-13-00585-f004:**
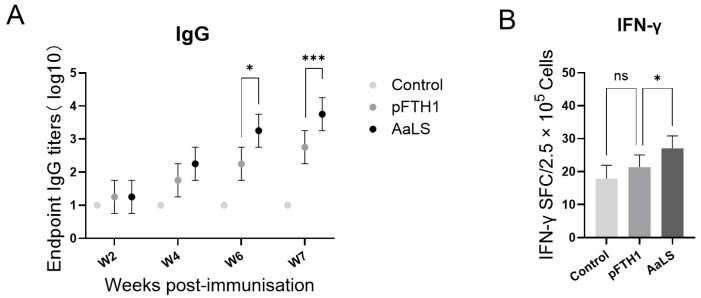
Immune responses in pigs following pFTH1 injection. (**A**) IgG antibody titers in pigs following injection with the control, pFTH1 or AaLS. (**B**) Number of IFN-γ+ cells in pigs following injection with the control, pFTH1 or AaLS (shown as mean ± SD, * *p* < 0.05, and *** *p* < 0.001).

**Figure 5 vaccines-13-00585-f005:**
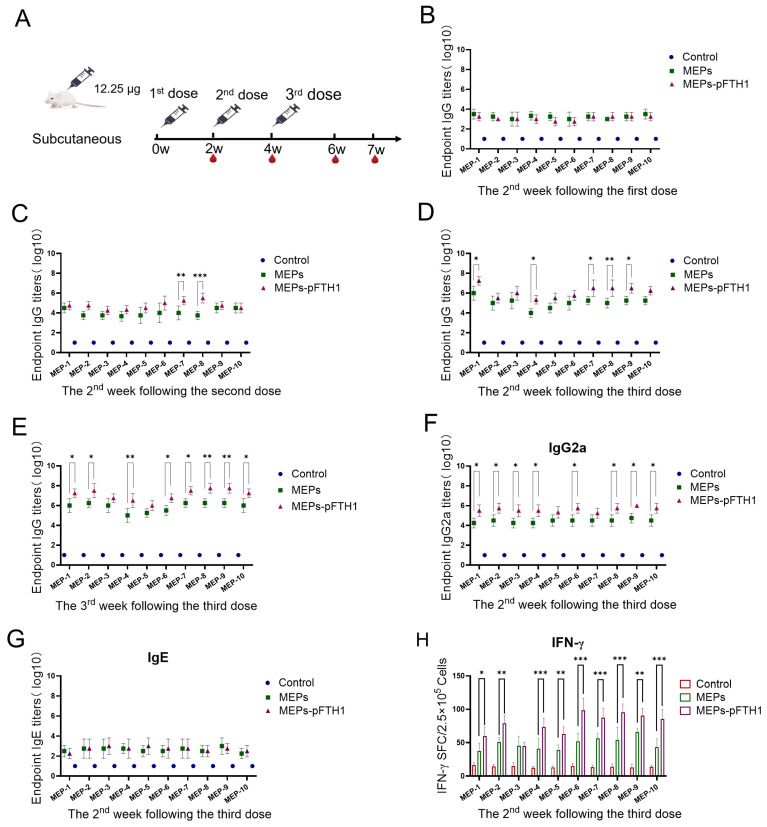
MEPs-pFTH1 nanoparticles induce robust humoral and cellular immune responses in BALB/c mice. (**A**) Immunization and blood sampling schedule. (**B**) Serum IgG titers in mice in the 2nd week following the first immunization. (**C**) Serum IgG titers in mice in the 2nd week following the second dose. (**D**,**E**) Serum IgG titers in mice in the 2nd and 3rd week following the third dose. (**F**,**G**) Serum IgG2a and IgE titers in mice in the 2nd week following the third dose. (**H**) Number of IFN-γ+ secreting splenic lymphocytes in the 2nd week following the third dose. (shown as mean ± SD, * *p* < 0.05, ** *p* < 0.01 and *** *p* < 0.001).

**Figure 6 vaccines-13-00585-f006:**
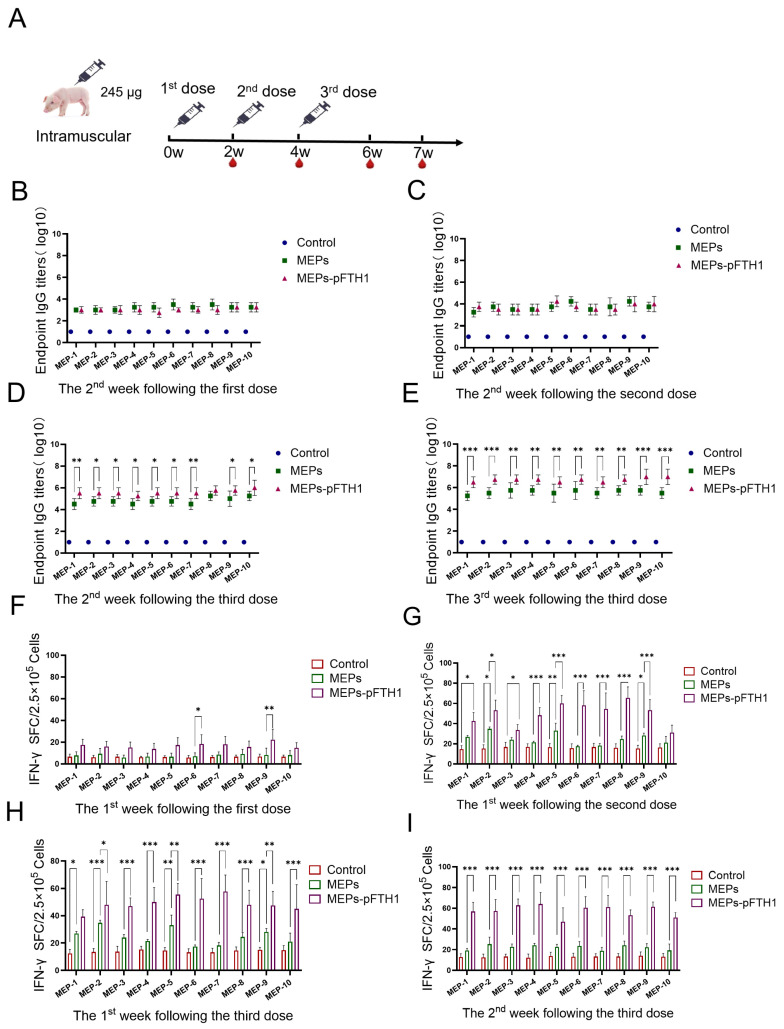
MEPs-pFTH1 nanoparticles induce potent humoral and cellular immune responses in pigs. (**A**) Immunization and blood sampling schedule. (**B**) Serum IgG titers in pigs in the 2nd week following the first dose. (**C**) Serum IgG titers in pigs in the 2nd week following the second dose. (**D**,**E**) Serum IgG titers in pigs in the 2nd and 3rd week following the third dose. (**F**,**I**) Number of IFN-γ-secreting PBMCs after restimulation with MEPs monomers in vitro at 1st week following the first dose (**F**) At the 1st week following the second dose (**G**), and 1st (**H**) and 2nd (**I**) week following the third dose. (shown as mean ± SD, * *p* < 0.05, ** *p* < 0.01 and *** *p* < 0.001).

**Figure 7 vaccines-13-00585-f007:**
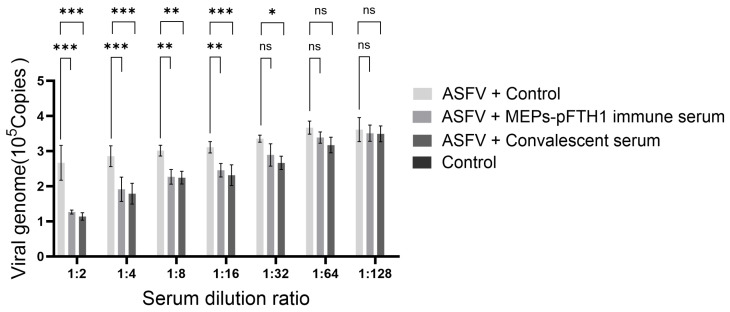
The sera of pigs immunized with MEPs-pFTH1 nanoparticle antigen can significantly inhibit the proliferation of ASFV. qPCR detection of viral genome copy number in PAMs treated by ASFV + MEPs-pFTH1 immune serum, or ASFV + convalescent serum (N = 3), (Data are expressed as mean ± SD, ns *p* > 0.05, * *p* < 0.05, ** *p* < 0.01 and *** *p* < 0.001).

**Table 1 vaccines-13-00585-t001:** Hematology parameters in immunized pigs.

Parameter	Control	pFTH1	MEPs	MEPs-pFTH1	Normal Reference Range
WBC (10^9^/L)	16.58 ± 1.09	16.96 ± 1.54	17.58 ± 0.95	18.55 ± 1.27	7.00–20.00
Neu%	39.93 ± 2.35	40.74 ± 1.09	38.70 ± 2.21	38.34 ± 1.17	23.00–64.00
Lym (%)	45.77 ± 3.42	45.20 ± 1.45	46.24 ± 1.82	46.66 ± 1.60	26.00–67.00
Mon%	7.03 ± 1.07	6.27 ± 0.80	7.83 ± 1.55	6.07 ± 1.61	2.00–11.00
Eos%	6.47 ± 1.70	6.93 ± 1.76	6.47 ± 0.74	8.13 ± 1.05	0.50–9.00
Bas%	0.08 ± 0.30	0.87 ± 1.21	0.77 ± 0.15	0.80 ± 0.36	0.00–1.00
Neu (10^9^/L)	6.61 ± 0.31	6.91 ± 0.69	6.80 ± 0.42	6.04 ± 1.24	2.00–7.00
Lym (10^9^/L)	7.59 ± 0.80	7.65 ± 0.46	8.13 ± 0.45	9.74 ± 2.04	7.00–10.00
Mon (10^9^/L)	1.17 ± 0.19	1.07 ± 0.20	1.39 ± 0.35	1.11 ± 0.23	0.12–1.20
Eos (10^9^/L)	1.08 ± 0.35	1.19 ± 0.36	1.14 ± 0.12	1.50 ± 0.10	0.02–0.50
Bas (10^9^/L)	0.13 ± 0.04	0.15 ± 0.04	0.14 ± 0.03	0.15 ± 0.06	0.00–0.10
RBC (10^12^/L)	8.07 ± 2.02	6.80 ± 0.73	7.18 ± 0.95	7.51 ± 0.74	6.00–8.50
HGB (g/L)	151.67 ± 20.31	131.67 ± 15.37	119.67 ± 6.66	119.33 ± 6.81	110.00–160.00
HCT (%)	47.23 ± 1.85	43.33 ± 4.21	40.37 ± 1.96	40.27 ± 1.76	37.00–54.00
MCV (fL)	56.00 ± 10.65	63.83 ± 0.70	56.93 ± 2.30	54.03 ± 7.58	80.00–100.00
MCH (pg)	17.77 ± 2.72	19.33 ± 0.58	16.87 ± 2.75	15.97 ± 2.47	27.00–34.00
MCHC (g/L)	320.20 ± 33.06	303.00 ± 9.85	296.00 ± 5.20	295.67 ± 4.93	320.00–360.00
RDW-CV (10^9^/L)	17.63 ± 1.72	16.07 ± 0.55	17.93 ± 1.93	18.23 ± 1.62	11.00–16.00
RDW-SD (fL)	34.27 ± 3.98	36.57 ± 0.80	35.83 ± 2.30	34.90 ± 2.88	35.00–56.00
PLT (10^9^/L)	303.00 ± 28.05	307.33 ± 50.36	305.00 ± 80.22	302.00 ± 37.24	100.00–400.00
MPV (fL)	7. 97 ± 0.95	7.97 ± 1.10	7.33 ± 0.61	7.37 ± 0.67	6.50–12.00
PDW (%)	16.17 ± 1.12	17.47 ± 0.49	16.43 ± 0.67	16.03 ± 0.49	15.00–17.00
PCT (%)	0.24 ± 0.10	0.27 ± 0.04	0.22 ± 0.07	0.20 ± 0.04	0.11–0.29

Note: There was no significant difference between groups (*p* > 0.05), and all hematology parameters in immunized pigs were within the normal reference ranges.

**Table 2 vaccines-13-00585-t002:** Serum biochemistry parameters.

Parameter	Control	pFTH1	MEPs	MEP-pFTH1	Normal Reference Range
TP (g/L)	65.67 ± 2.36	63.67 ± 4.25	65.00 ± 2.50	63.33 ± 3.25	60.00–80.00
ALB (g/L)	34.57 ± 1.78	33.40 ± 2.54	33.70 ± 1.93	32.23 ± 1.22	18.00–38.00
GLO (g/L)	31.10 ± 1.51	30.27 ± 1.72	31.30 ± 0.69	31.77 ± 1.79	22.00–62.00
A/G	1.11 ± 0.08	1.10 ± 0.02	1.08 ± 0.05	1.02 ± 0.02	1.50–2.50
TBIL (μmol/L)	3.53 ± 1.03	3.70 ± 0.43	3.72 ± 0.85	3.44 ± 0.79	0.00–17.10
ALT (U/L)	36.33 ± 4.04	36.67 ± 2.08	37.33 ± 2.08	35.67 ± 3.51	9.00–58.00
AST (U/L)	34.33 ± 9.02	36.33 ± 2.08	36.00 ± 3.61	38.33 ± 3.06	16.00–84.00
AST/ALT	0.90± 0.20	1.00 ± 0.11	0.96 ± 0.06	0.79 ± 0.01	0.80–1.50
GGT (U/L)	42.83 ± 3.19	41.80 ± 3.20	44.30 ± 4.50	42.13 ± 3.43	31.00–52.00
ALP (U/L)	82.67 ± 13.58	81.00 ± 17.06	83.33 ± 12.06	85.33 ± 18.61	41.00–176.10
TBA (μmol/L)	0.79 ± 0.32	0.86 ± 0.40	0.81 ± 0.19	0.77 ± 0.22	0.00–15.00
CK (U/L)	301.33 ± 27.79	314.67 ± 43.84	287.67 ± 18.15	296.67 ± 27.06	50.00–689.40
AMY (U/L)	1823.67 ± 564.1	1906.67 ± 291.5	1934.00 ± 190.31	1858.62 ± 239.55	43.50–176.00
TG (mmol/L)	0.66 ± 0.11	0.82 ± 0.10	0.81 ± 0.09	0.76 ± 0.01	0.46–0.94
CHOL (mmol/L)	2.35 ± 0.57	2.34 ± 0.45	1.95 ± 0.08	2.10 ± 0.24	1.30–3.60
GLU (mmol/L)	5.57 ± 1.06	5.46 ± 0.67	4.92 ± 0.91	5.06 ± 0.27	4.72–8.89
CRE (μmol/L)	108.67 ± 18.50	111.00 ± 24.58	106.33 ± 16.17	108.67 ± 9.61	44.00–186.00
BUN (mmol/L)	4.30 ± 0.24	4.23 ± 0.24	4.42 ± 0.31	4.56 ± 0.41	2.10–10.70
tCO2 (mmol/L)	20.33 ± 0.58	20.33 ± 1.53	20.00 ± 1.00	20.00 ± 1.73	17.00–28.00
Ca (mmol/L)	2.48 ± 0.09	2.46 ± 0.11	2.28 ± 0.27	2.45 ± 0.14	1.63–2.90
P (mmol/L)	1.60 ± 0.09	1.62 ± 0.23	1.43 ± 0.36	1.41 ± 0.26	1.16–3.55

Note: There was no significant difference between groups (*p* > 0.05), and all serum biochemistry parameters in immunized pigs were within the normal reference ranges.

## Data Availability

The authors confirm that the data supporting the findings of this study are available within the article.

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
