# Peer review of "A Multiepitope Nanovaccine Candidate Adjuvanted with Porcine Ferritin Scaffold for African Swine Fever Virus"

_vaccines, 2025, doi:10.3390/vaccines13060585_

Round 1
Reviewer 1 Report (Previous Reviewer 1)
Comments and Suggestions for Authors
Western blot data: Along with the SDS-PAGE results, please consider including Western blot data for all purified proteins prior to their incorporation into nanoparticles.
ELISA and ELISPOT: Since neither the virus nor its components were used, the data appear to reflect the immune response to the nanoparticles themselves rather than to ASFV.
Convalescent serum: Please specify the origin of the convalescent serum used in the experiments.
VN assay: How were the observed effects—such as cell aggregation, cellular disruption, and other signs of CPE—characterized in the experimental versus control groups?
Was there a serum dilution that completely inhibited these effects?
ASFV strain: Indicate which ASFV strain was used. Do you believe that all 12 strains included in the bioinformatic analysis should be tested to evaluate whether the vaccine induces a broad immune response?
qPCR: A reference or detailed protocol for the qPCR assay is needed.
ELISPOT stimulation: Please clarify why 1 μg/mL MEP monomer was used to stimulate lymphocytes in the ELISPOT assay instead of individual proteins.
Immune response evaluation: What method was used to evaluate the immune response to individual proteins?
Author Response
Dear Reviewer:
Thank you for the reviewers’ comments, these comments are extremely valuable and insightful for revising and improving our manuscript, as well as for guiding our research. We have carefully reviewed the feedback and made thorough revisions to the manuscript according to the reviewers’ suggestions. The corrections and responses to the reviewer’s comments are shown below:
Reviewer #1:
Comments 1: Western blot data: Along with the SDS-PAGE results, please consider including Western blot data for all purified proteins prior to their incorporation into nanoparticle
Response 1: We have added the Western blot data for all purified proteins prior to their incorporation into nanoparticle in the revised manuscript according to the Reviewer’s comment.
Comments 2: ELISA and ELISPOT: Since neither the virus nor its components were used, the data appear to reflect the immune response to the nanoparticles themselves rather than to ASFV.
Response 2: Western blot (WB) experiment revealed that the recombinant antigen reacted with specific antibodies in ASFV-positive serum. The results indicate that the recombinant antigen has ASFV antigenicity.
Comments 3: Convalescent serum: Please specify the origin of the convalescent serum used in the experiments.
Response 3: The convalescent serum used in this study was collected from pigs naturally infected by ASFV.
Comments 4: VN assay: How were the observed effects—such as cell aggregation, cellular disruption, and other signs of CPE—characterized in the experimental versus control groups?
Response 4: The cell pathological changes can be observed under the bright field of the microscope.
Comments 5: Was there a serum dilution that completely inhibited these effects?
Response 5: In this study, the immune serum was diluted 2-fold and added to porcine alveolar macrophages infected with 0.1 MOI ASFV. The virus remained detectable by qPCR, but its cytopathic effect was significantly dampened compared to the virus only group. Further investigation involving reduced viral dosages may reveal a phenomenon of complete inhibition. Nevertheless, the current study has unequivocally demonstrated that the immune serum exerts an inhibitory effect on the virus.
Comments 6: ASFV strain: Indicate which ASFV strain was used. Do you believe that all 12 strains included in the bioinformatic analysis should be tested to evaluate whether the vaccine induces a broad immune response?
Response 6: This study selected the strains that were prevalent in China in 2019. During the antigen design stage, the common highly conserved proteins of 12 strains that have been prevalent in the past decade were selected, and high-affinity epitopes were screened from these conserved proteins. A recombinant multi-epitope antigen was constructed through artificial synthesis. In theory, this antigen should have the ability to induce a broad immune response. However, due to the difficulty of culturing African swine fever virus in vitro and the requirement that related experiments be conducted in a biosafety level 3 (BSL-3) laboratory, it is an extremely challenging task to obtain these 12 strains and comprehensively evaluate whether the vaccine can induce a broad immune response due to resource limitations, and it is difficult to achieve in the short term. Nevertheless, we highly agree with your suggestion and recognize its significant value. In future in-depth studies, if conditions permit, we will further explore this direction.
Comments 7: qPCR: A reference or detailed protocol for the qPCR assay is needed.
Response 7: We have added the detailed protocol for the qPCR assay into the revised manuscript in response to the Reviewer’s comment.
Comments 8: ELISPOT stimulation: Please clarify why 1 μg/mL MEP monomer was used to stimulate lymphocytes in the ELISPOT assay instead of individual proteins.
Response 8: In the ELISPOT stimulation experiment, the individual MEP proteins we used stimulate the lymphocytes of immune animals respectively. To distinguish the individual proteins from nanoparticles, they are described as MEP monomers in the text.
Comments 9: Immune response evaluation: What method was used to evaluate the immune response to individual proteins?
Response 9: We evaluated the ASFV antigenicity of individual proteins by the WB method. Mice and pigs were immunized with the MEP monomer proteins mixture, and the immune response induced by individual proteins were evaluated by detecting IgG, IgG2a, IgE, and IFN-γ in the serum.
We have made effort to improve and revise the manuscript. The changes (marked in red) do not affect the overall content or structure of the manuscript.
Sincerely
Lidan Sun
Sunlidan0399@163.com
Reviewer 2 Report (New Reviewer)
Comments and Suggestions for Authors
Due to the global spread of the ASF virus, improvements in prevention and control of the virus, as well as the development of a safe and effective ASF vaccine, are essential. It is therefore necessary to identify the ASFV protective proteins as well as the T- and B-epitopes in these proteins. Such knowledge is very important for the development of a safe vaccine against ASF. Bioinformatic methods serve as an effective strategy to identify T- and B-epitopes and create vaccine candidates for humans and animals. The authors of this manuscript developed a novel MEPs-pFTH1 self-assembling candidate nanovaccine that induces humoral and cellular immune responses in both pigs and mice. Although the presented data indeed demonstrate that immunization with both the MEPs-pFTH1 and MEPs monomers induces humoral and T-cell immune responses, the protective properties of this vaccine candidate remain unclear. A large number of published data indicate that the induction of an immune response in pigs often does not correlate with the development of protection. Therefore, a classical experiment on immunization of susceptible animals with subsequent challenge with the ASFV is needed to discuss the possibility of using the proposed approach to develop a vaccine against ASF.
- It should be noted that the authors do not indicate whether similar studies have been conducted previously, do not discuss whether their data are consistent with the data of other researchers. This should be added to the Discussion section.
- Previously, the authors in 2023 published a similar article on the development of nanovaccines against ASF, which was also not tested for protective activity. The lack of understanding of the protective properties of the developed vaccine candidates significantly reduces the value of these studies and is a significant limitation. This should be reflected in the text of the manuscript.
- Lines 135-136. Check for an error in the temperature.
Despite the indicated comments, the article is interesting and can be published after the text of the manuscript is corrected.
Author Response
Dear Reviewer:
Thank you for the reviewers’ comments,These comments are extremely valuable and insightful for revising and improving our manuscript, as well as for guiding our research. We have carefully reviewed the feedback and made thorough revisions to the manuscript according to the reviewers’ suggestions. The corrections and responses to the reviewer’s comments are shown below:
Comments1. It should be noted that the authors do not indicate whether similar studies have been conducted previously, do not discuss whether their data are consistent with the data of other researchers. This should be added to the Discussion section.
Response 1: The discussion of the manuscript mentions similar studies, such as those utilizing ferritin and AaLS self-assembled nanoparticle antigens, constructed through immunoimmune informatics tools to predict T and B cell epitopes of African swine fever virus (ASFV) target antigens, successfully induced robust humoral and cellular immune responses. But none of these studies conducted an animal challenge test.
Comments 2. Previously, the authors in 2023 published a similar article on the development of nanovaccines against ASF, which was also not tested for protective activity. The lack of understanding of the protective properties of the developed vaccine candidates significantly reduces the value of these studies and is a significant limitation. This should be reflected in the text of the manuscript.
Response 2: The article published in 2023 was expounded in the Introduction of the manuscript. Due to the requirement for an Animal Biosafety Level 3 (ABSL-3) facility, we were unable to perform a virus challenge test to assess the protective effect of the vaccine. However, we demonstrated the potential protective effect of the vaccine by showing that sera from pigs vaccinated with the nanoparticle vaccine significantly inhibited ASFV replication in vitro in the serum micro-neutralization assay. But it is also listed as a research limitation in the discussion part of the manuscript.
Comments 3. Lines 135-136. Check for an error in the temperature.
Response 3: We apologize for overlooking this detail, we have made the corresponding correction.
- Despite the indicated comments, the article is interesting and can be published after the text of the manuscript is corrected.
Response 4: Thanks to the reviewers for their recognition of this manuscript and for providing valuable suggestions.
We have made effort to improve and revise the manuscript. The changes (marked in red) do not affect the overall content or structure of the manuscript.
Sincerely
Lidan Sun
Sunlidan0399@163.com
Round 2
Reviewer 1 Report (Previous Reviewer 1)
Comments and Suggestions for Authors
All the comments have been addressed.
Author Response
Dear Reviewer,
Your insightful comments and constructive feedback have been instrumental in elevating the manuscript to meet rigorous academic standards. The expert guidance provided has significantly enhanced both the scientific validity and structural coherence of this work. We wish to express our deepest gratitude for your meticulous review and valuable suggestions, which have profoundly strengthened the scholarly contribution of this research. Please accept our sincere appreciation for your dedication to advancing the quality of academic discourse.
Sincerely,
Lidan Sun
sunlidan0399@163.com
Reviewer 2 Report (New Reviewer)
Comments and Suggestions for Authors
In my opinion, the manuscript can be published despite the lack of evidence of protective activity. The results of the serum micro-neutralization assay often do not agree with the results of in vivo experiments, we have repeatedly observed this in our work. I hope that in the future, the authors will test their prototypes in immunization/challenge experiments with pigs.
Author Response
Dear Reviewer,
Your insightful comments and constructive feedback have been instrumental in elevating the manuscript to meet rigorous academic standards. In future work, if we have the opportunity, we will evaluate the protective efficacy of this vaccine. We wish to express our deepest gratitude for your meticulous review and valuable suggestions, which have profoundly strengthened the scholarly contribution of this research. Please accept our sincere appreciation for your dedication to advancing the quality of academic discourse.
Sincerely,
Lidan Sun
sunlidan0399@163.com
This manuscript is a resubmission of an earlier submission. The following is a list of the peer review reports and author responses from that submission.
Round 1
Reviewer 1 Report
Comments and Suggestions for Authors
The overall idea of this manuscript is interesting and worth exploring. However, based on my understanding, when combining multiple peptides or immunogens, it is expected that the immune responses to each component be evaluated individually. Additionally, since the epitopes were selected after screening multiple genotypes/strains, their ability to induce a broad immune response should be further addressed.
Another concern is that the majority of the in vitro assay data is not convincing because neither live virus nor a well-characterized viral component was used. Probably, a commercial ELISA should have been performed to demonstrate that serum from vaccinated animals can recognize the target antigen.
Specific Comments:
- Spell out all abbreviations when first used.
- Italicize strain names.
- Ensure consistent font size throughout the manuscript.
- SDS data: Information on nanoparticles is missing.
- ELISA and ELISPOT: Since neither the virus nor its components were used, the data appears to show the immune response to nanoparticles rather than to ASFV.
VN test: ASFV was used, and it appears to have caused CPE in PAMs. Because of that there are a couple of questions
- Specify the type of convalescent serum used.
- Was there a serum dilution that completely blocked this effect?
- Indicate which virus was used.
- Have you tested a more diluted virus (MOI 0.01)?
- Consider including a virus control for your VN data.
- Additional qPCR data is needed. Provide details on the number of replicates performed for each sample/dilution.
Reviewer 2 Report
Comments and Suggestions for Authors
This article presents an interesting preclinical study of a novel prototype vaccine against ASFV-associated infection, which comprises a conjugate of two recombinant components: a ferritin scaffold and polyepitope cassettes. The authors have shown the vaccine safety and immunogenicity in both mouse and pig models; in addition, it induced the production of neutralizing antiviral antibodies. The paper is logically given and argued, and the results are well-illustrated, so it is of obvious value for the development of new ASFV vaccines, especially in view of their scarcity. However, the manuscript contains some flaws that should be corrected to make it suitable for publication.
First, the Title should be shortened by removing '84.7%...'. In addition to the Title, this value is also annoyingly repeated several times in the text (in lines 21, 254, 437, 455, 464, etc.)
TEPs are not deciphered in the Abstract. And why exactly were T-cell epitopes taken as a vaccine basis if immunogenicity was also assessed by antibody production?
The Discussion is quiet fine, but lines 80-83 seem to be the beginning of subsection; they are poorly connected to the previous text and should be referenced.
Line 88: not only MEPs-pFTH1, but also monomeric MEPs were found to be immunogenic.
The Materials and Methods Section requires more refinements. The sources of sequences and epidemiological data should be clearly specified (not only as NCBI home page, lines 95-96).
The parameters of TEP prediction (proteolysis sites number, immunogenicity score and threshold, MHCI alleles, etc.) should be indicated.
Line 100: NetMHCpanEL 4.1 should be referenced.
From lines 108-109 and 117-118, it's not clear how were the expression constructs organized, so these sentences need to be rephrased. Moreover, 'restriction sites' mentioned here should be described.
Were the (DE3) E.Coli strains used in the study? The strain names should be corrected in lines 113 and 123.
The authors stated the benefits of the recombinant ferritin scaffold, but the native structure of the expressed pFTH1 was not validated in the study even by the WB with the anti-ferritin antibodies.
Line 136: are molar or mass ratios indicated?
Line 142: an exclusion chromatography features (resin type, flow rate, buffers, etc.) should be given.
Line 145: what was the protein load (µg per lane) in the samples?
'Transfer apparatus' in line 148 and 'chemiluminescence' in lines 153-154 should be specified.
Mice breed should be indicated in line 163. By the way, the piglets (line 228) and isolation/maintaining of piglet-derived alveolar macrophages are not described in Materials and Methods at all.
“Guide for the Breeding and Use of Laboratory Animals” mentioned in lines 164-165 should be referenced.
Line 169: four mice per group is a very small sample size. This should be mentioned in the Discussion as a study limitation.
Why were mouse IgG, IgG2a and IgE detected separately (line 174)? And why it was not performed for pigs? If only because the secondary anti-pig Abs were unavailable, it should be indicated as another study limitation.
Line 200: 'various dilutions' need to be specified, as well as 'MEP monomer stimulator' in line 216.
Which exactly detection antibody (line 219) was used?
Line 234: how was ASFV previously titrated to calculate MOI? Moreover, how the virus was propagated and which viral strain was used?
The authors claimed that they were unable to perform any challenge studies because of BSL3 facility requirements (lines 458-460). But how then were the neutralization tests conducted? They implies the live virus usage, and the authors didn't state that these procedures were performed in BSL3 lab.
Line 241: the viral DNA could not be purified by mild centrifugation only. The DNA purification protocol as well as qPCR details (detection system, primers, machine, etc.) should be given here.
How was the data normality checked prior to T-tests (line 245)?
The 'Results' subtitle is missing.
All the Figures should be enlarged.
Figure 1 shows that pFTH1 self-assemblies spontaneously to form 24-mer. But only the monomeric pFTH1 (~39 kDa) can be seen at all SDS-PAGE results given in Figure 3B. The addition of any reducing reagents (bME, DTT, etc.) in the samples was not stated by the authors. And by the way, were the monomeric MEPs self-organized in the structure shown in Figure 1B?
Figure 2 is virtually unreadable. What was the principle of each MEP compilation?
Lines 276-278: renaturation protocol should be given in detail in Materials and Methods.
The table (+/-) above Figure 3B is incomprehensible and inconsistent with the SDS-PAGE results. In addition, Figure 3C shows that monomeric MEPs were also present in each MEP-pFTH1 conjugate. Were these results obtained after gel-filtration of the conjugates? If yes, its incorrect to compare these preparations effects to those for monomeric MEPs. If no, WB results should be shown for completely purified MEPs-pFTH1.
AaLS (Aquifex aeolicus lumazine synthase) scaffold (line 305) obtaining or source should be described in Materials and Methods. Why was it used as an alternative carrier protein?
Which 'specific antibodies' and 'anti-scaffold antibodies' are implied in lines 306-307 and 308, respectively?
Line 315: how many weeks post-immunization was the number of IFNg+ T-cells assessed?
Line 321: to state that immunization has no impact on the animals, it's important not only to compare the values with 'normal reference ranges', but also to assess the intergroup differences. Although it stated that 'no significant differences in serum chemistry parameters' were found among the groups, p-values are not given in Tables 1 and 2. By the way, 'serum biochemistry' would be a more correct term.
In Figures 5A and 6A, doses should be indicated.
Line 342: why the responses were evaluated at 3 weeks after the third dose, if the sampling regimen described in Materials and Methods implies analysis in 2 weeks after each dose?
Lines 352 and 370: 'control' (PBS or pFTH1) should be clearly specified. Moreover, control group results should be added to plots 5B-G (e.g., there are controls in analogous Figure 6B-E).
Why T-cell responses in mice were evaluated at two weeks after the third dose (Figure 5H) while maximal IgG titers were revealed at three weeks after the third dose (Figure 5E)?
The content and description of Figure 6I are inconsistent (2nd week vs 3rd week).
The morphological changes illustrated by Figure 7A seem unconvincing. It would be better to give the ICC results using any anti-ASFV mAbs. By the way, a standard serum virus neutralization assay should be conducted with the same samples side-by-side with the developed assay.
Figure 7B lacks the background control (w/o ASFV).
The Discussion needs significant refinement. Lines 412-417 and 420-425 should be referenced. Instead of the repetitive description of authors' own results (lines 435-449) already presented in the Conclusion and Abstract, it's necessary to add a detailed discussion of other papers (not only the authors' previous work [43]) on the development of ASFV vaccines, on the use of polyepitope protein constructions and ferritin as a carrier protein, as well as a paragraph on study limitations. The sudden mention of SARS-CoV-2 T-cell epitope vaccine (lines 426-431) seems illogical.
The T-cell responses revealed in this work cannot be recognized as Th1-dominant (line 465) based on IFNg secretion alone without analysis of the expression of other cytokines by the specific T-cells.
Finally, there are some language flaws, word duplications and typos in the article (e.g., in lines 41, 120, 122, 259, 289, etc.), so moderate English editing is recommended to the authors.